# Association between Sense of Loneliness and Quality of Life in Older Adults with Multimorbidity

**DOI:** 10.3390/ijerph20032615

**Published:** 2023-02-01

**Authors:** Anna Vespa, Roberta Spatuzzi, Paolo Fabbietti, Mirko Di Rosa, Anna Rita Bonfigli, Andrea Corsonello, Pisana Gattafoni, Maria Velia Giulietti

**Affiliations:** 1Scientific and Technological Area, Department of Neurology, Italian National Research Center On Aging (IRCCS INRCA), 60124 Ancona, Italy; 2Department of Mental Health, ASP Basilicata, 85100 Potenza, Italy; 3Laboratory of Biostatistics, Italian National Research Center On Aging (IRCCS INRCA), 60124 Ancona, Italy; 4Scientific Direction, Italian National Research Center On Aging (IRCCS INRCA), 60124 Ancona, Italy; 5Unit of Geriatric Pharmacoepidemiology and Biostatistics, Italian National Research Center On Aging (IRCCS INRCA), 87100 Cosenza, Italy; 6Clinic of Internal Medicine and Geriatric, Italian National Research Center On Aging (IRCCS INRCA), 60124 Ancona, Italy; 7Department of Neurology, Italian National Research Center On Aging (IRCCS INRCA), 60124 Ancona, Italy

**Keywords:** sense of loneliness, social support, quality of life, older adults, multimorbidity

## Abstract

Background: Multimorbidity has been associated with adverse health outcomes, such as reduced physical function, poor quality-of-life (QoL), poor self-rated health. Objective: The association between quality of life, social support, sense of loneliness and sex and age in older adult patients affected by two or more chronic diseases (multimorbidity) was evaluated. Methods: Patients n. 162 with multimorbidity and living with family members. Tests: MMSE-Mini-Mental-State-Examination; ADL-Activities of Daily Living; Social Schedule: demographic variables; Loneliness Scale -de Jong Gierveld; Quality-of-Life-FACT-G; WHOQOL-BRIEF Social relationships. Statistical analysis: Multivariate Regression Analysis. Results: The patients with three or more diseases have worse dimensions of FACT-G total score (*p* = 0.029), QoL Physical-well-being (*p* = 0.003), Social well-being (*p* = 0.003), Emotional-well-being (*p* = 0.012), Functional-well-being (*p* < 0.001), than those with two. Multiple linear regression QoL: FACT_G total score, PWB, SWB, EWB, FWB as dependent variables. In the presence of multimorbidity with an increase in the patient’s age FACT-G total score (B = −0.004, *p* = 0.482), PWB (B = −0.024, *p* = 0.014), SWB (B = −0.022, *p* = 0.051), EWB (B = −0.001, *p* = 0.939), FWB (B = −0.023, *p* = 0.013) decrease by an average of 0.1, and as the sense of solitude increases FACT-G total score (B = −0.285, *p* < 0.000), PWB (B = −0.435, *p* < 0.000), SWB(B = −0.401, *p* < 0.000), EWB(B = −0.494, *p* < 0.000), FWB(B = −0.429, *p* < 0.000) decrease by 0.4. Conclusions: A sense of loneliness and advancing age are associated with bad quality-of life in self-sufficient elderly patients with multimorbidity. Implications for Practice: Demonstrating that loneliness, as well as in the presence of interpersonal relations, is predictive of worse quality of life in patients with multimorbidity helps identify people most at risk for common symptoms and lays the groundwork for research concerning both diagnosis and treatment.

## 1. Introduction

Multimorbidity is a major concern for health services, health research, and health policy [1,2,3,4]. As a result of increasing longevity, multiple comorbid conditions, commonly referred to as ‘multimorbidity’, defined as two or more chronic long-term chronic conditions, have also become progressively more common among older adults [1,5,6,7,8,9]. Health determinants include medical and genetic conditions, frailty, etc. (intrinsic determinants) [10]. On the other hand, the physical and social environment (extrinsic determinants) are grouped under the concept of “social vulnerability” [8,11,12,13]. Social vulnerability has been associated with a higher prevalence of frailty and higher levels of hospital mortality [2,3,12,14,15,16,17,18,19,20,21,22,23,24,25,26].

Some studies have demonstrated the exponential increase in disability with increasing numbers of chronic diseases and the effects of multimorbidity (absolute number of diseases or diseases combinations) on health outcomes such as physical functioning, quality of life, or mental health [1,11,16,19,20,24,26,27]. In this context many studies have highlighted the importance of the quality of social relations to protect from the development and progression of physical illnesses and to promote mental health (particularly in the elderly). Loneliness or perceived absence of positive social relationships has been linked to lower longevity, particularly among older adults [5,7,22,23,28,29,30,31,32,33,34].

In fact, older adults can suffer events such as illness, widowhood, and reduced mobility as well as a decline in economic resources, and all of this can lead to an increased risk of social isolation [16]. Social isolation indicates an objective state of minimal social contact with other individuals, while solitude reflects a subjective state of lack of the desired affection and closeness to a significant other or intimate (e.g., emotional loneliness) or close friends and family members (e.g., relational solitude). Furthermore, loneliness and living alone are related but not overlapping categories [13,35,36,37,38,39]. Studies on older adults show that living alone is not necessarily indicative of loneliness [40]: people living alone reporting frequent social contacts and active social involvement in community organizations [36,41,42,43]. Thus, researchers have distinguished loneliness from the experience of being alone or solitary. Loneliness has an involuntary and ego-dystonic character and this includes the desire for human contact (i.e., belonging and/or intimacy). This desire is one of the aspects that differentiate involuntary loneliness [loneliness] from voluntary loneliness (loneliness) [13,44,45].

In old age, perceived social isolation, and a sense of loneliness are also major risk factors for broad-based morbidity (both psychological and physical) [14,28]. Loneliness has been shown to prospectively predict increased depressive symptomatology, impaired cognitive performance, dementia progression, significant likelihood of nursing home admission, and multiple disease outcomes with functional limitations in older people (e.g., hypertension, heart disease, and stroke in older persons) [16,20,40,46,47,48,49]. Thus, a sense of social isolation was recognized as a significant risk factor for morbidity and mortality [5,18,20,22,30,31,34,50,51]. Although more attention has recently been paid to this phenomenon [16], relatively little is known about the association between physical multimorbidity and the sense of loneliness [5,18] as more and more elderly people live alone, and some of them are at risk of feeling lonely or socially isolated [30,32].

Limited empirical data exists on the impact of loneliness on mortality as well as on mechanisms. Some authors suggest that materialist models of multimorbidity and functional limitation at older age cannot, on their own, explain the health inequalities as the behavioral and psycho-social factors play an important role.

Thus, the present study examines what affects emotional state and quality of life (health-related quality of life—HRQoL), when two or more physical illnesses (multimorbidity) are present in an elderly person, and whether a sense of loneliness plays a role. To isolate the sense of loneliness, elderly patients with multimorbidity, who live in a family (and therefore with the objective presence of certain relationships) were studied. The study of this association can help promote targeted multidimensional prevention interventions, focused on the quality of the relationship, aimed at favoring the maintenance or restoration of a good quality of life in elderly people with multimorbidity [3,6,8].

## 2. Materials and Methods

This study uses data from the Multidimensional Evaluation Form in Geriatrics project, a prospective observational study aimed at collecting data about the patterns and quality of prescriptions among older patients admitted to acute care wards of geriatric medicine located in Central Italy. The study was approved by the ethics committees of the INRCA-IRCCS National Institute of Science and Health for Aging in Ancona, Italy (IRB approval number: Ethics Committee IRB CdB:18019; IRB CdB:16025). Multimorbidity is defined as the simultaneous presence of two or more chronic physical health conditions. For this analysis, patients affected by two or more (three to five) chronic health diseases were included: malign tumors, cardiovascular diseases, chronic pulmonary diseases, diabetes mellitus, hypertension (Table 1). All of these conditions were assessed by diagnosis of the disease carried out by the doctors.

Additionally, the use of treatment/medication received in the 12 months prior to interview was indicative of a diagnosis and was included in prevalence estimates for each disease.

### 2.1. Participants

Two hundreds ninety three elderly patients affected by 2 and ≥3 different diseases were consecutively recruited and asked to participate during follow-up medical visits at the Departments of INRCA, National Institute of Science and Health for Aging. Only two hundred and sixty eight patients joined the study. Thirty-eight patients did not meet the inclusion criteria; they lived alone. Fourteen patients did not answer all of the questionnaires: it was therefore decided not to consider them in the study. So, the sample included 216 patients. The demographic variables are described in Table 1. They compiled the expected tests administered by specifically trained psychologists. The diagnosed diseases were the following: malign tumors, cardiovascular diseases, diabetes, hypertension, pulmonary diseases. The presence of two or more pathologies in a single patient has been documented by the physicians (Table 1). Moreover, the patients showed a medium (7%) to good (93%) self- functional sufficiency evaluated by ADL (Activities of Daily Living) and intact cognitive abilities (no deterioration) evaluated by MMSE and verified during interview.

After completing the initial medical examination, they were referred to the investigator. Inclusion criteria included age (over 65), diagnosis of different diseases (from 1 to 4 years); presence of social relationships (they were living with spouses or sons/daughter or other relatives); having no diagnosis of neurological or cognitive impairments; being proficient in the Italian language; providing a written informed consent.

Patients were excluded if: were unable to provide informed consent; had other forms of disease (dementia, terminal illness); MMSE < 24; were not self-sufficient (ADL); were using any type of psychotropic drugs (including antidepressants); they were not living with relatives; financial difficulties.

Age and demographic data including marital status and educational levels were collected (Table 1).

### 2.2. Measures

All of the patients filled in the following tests:

(a) ADL—Physical functioning: activities of daily living. Limitation in ADL was used to assess physical functioning [52]. The questions were based on self- reported difficulty in engaging in activities during the last 30 days, using a five-point response scale ranging from none to extreme difficulty. The ADL measure included in SAGE was based on WHODAS 2.0 and has been validated in LMICs by WHO and collaborating agencies. WHODAS 2.0 is validated cross-culturally through a systematic research study. The cross-cultural applicability research study used various qualitative methods to explore the nature and practice of health status assessment in different cultures.

The study included linguistic analysis of health-related terminology, key informant interviews, focus groups, and quasi-quantitative methods such as pile sorting and concept mapping (carried out in tandem). Information was gathered on the conceptualization of disability and on important areas of day-to-day functioning. In this study, severe and extreme difficulties were combined to represent limitation in a particular activity. We have used an extended set of ADL that included sitting for long periods, walking 100 m, standing up, standing for long periods, climbing one flight of stairs, stooping/kneeling/crouching, picking up things with fingers, extending arms above shoulders, concentrating for 10 min, walking a long distance (1 km), bathing, getting dressed, carrying things, moving around inside home, getting up from lying down, and getting to and using the toilet. For the analysis, a dichotomous variable was created, which took a value of 1 if the respondent noted a limitation in one or more of the above ADLs (1+ ADL) and 0 otherwise.

(b) MMSE—Mini Mental State Examination [53] measures cognitive functions. The Italian versions of the MMSE modified from the Los Angeles Epidemiologic Catchment Area study was used. MMSE scores were dichotomized to indicate presence or absence of cognitive impairment using the published cut point of ≥24 (cognitive impairment absent) and <24 (cognitive impairment present).

(c) Social Schedule describing sex, age, marital status, educational levels, presence of different kinds of interpersonal relation. A self-evaluation of patient health was conducted, consisting of a single question: “How would you classify your health: excellent, good, acceptable, bad or very bad?” The score for “very bad” was 1 and for “excellent” it was 5. The patient was invited to select the most appropriate score. A schedule of clinical diagnosis (with years from first diagnosis) was included by physicians. Moreover, some questions were asked about the presence of relationships with family and friends (i.e., “who do you live with?”; How many times a week do you meet other family members or friends?)

(d) Quality of Life—Functional-Assessment of Cancer-Therapy-General (FACT-G) [54] has all requirements including reliability and validity for use in oncology clinical trials (Italian version). The score sums up to a total ranging from 0 to 108 points, where a higher score indicates better quality of life. It includes the following subscales: physical-well-being (PWB), social-well-being (SWB), emotional-well-being (EWB), functional-well-being (FWB), FACT-general-summary-score (FACT_G).

(e) Loneliness Scale by Jenny de Jong Gierveld [55,56] consisting of eleven items; six are formulated negatively and five are formulated positively. Loneliness is seen as a subjective experience and is, as such, not directly related to situational factors. The scale describing the sense of loneliness, or subjective social isolation, is defined as a situation experienced by the participant as one where there is an unpleasant or inadmissible lack of (quality of) certain relationships. Loneliness includes situations where the number of existing relationships is smaller than desirable or acceptable, as well as situations where the intimacy wished for has not been realized.

The11-item multidimensional scale is descriptive of the following dimensions: (1) severe feelings of loneliness as well as less intense loneliness feelings; (2) negative as well as positive items; and (3) a latent continuum of deprivation. In addition, the scale met the criteria of the dichotomous logistic Rasch model.

(f) The questionnaire of social relationships of WHOQOL-BRIEF was used. This is a validated Italian version by De Girolamo [57]. It is a 13-item questionnaire exploring dimensions such social relationship (three items), environment (eight items) and two global HRQoL items assessing an individual’s overall satisfaction with life and general sense of personal well-being. Responses to each item are coded from 1 to 5, summed, and transformed to a scale from 0 (worst HRQoL) to 100 (bestHRQoL).

### 2.3. Clinical Evaluation

The different diagnosis were made by the physicians for each diseases following the guide lines for each pathologies and in the occasion of follow-up visits at the time of test administration expected for this research.

## 3. Statistical Analysis

Data were expressed as means ± standard deviation (continuous variables) or as percentage (categorical variables). All continuous variables were checked for normality of distribution by the Kolmogorov–Smirnov test and the result was that all of them were normally distributed. The reliability of the FACT–G total score and its subscales was assessed by Cronbach’s coefficient alpha. The pathologies number was dichotomized by less than three and three or more pathologies. Statistical comparison between these two categories of pathologies (multimorbidity) and the quality of life was performed by t-Student test. The sense of loneliness was categorized into four groups (not loneliness, moderate loneliness, severe loneliness, very severe loneliness). The WHOQOL was dichotomize by no or few social relations and presence of good social relations. Pearson’s coefficient was used to assess correlations between studied variables. Multiple linear regression models were assessed to evaluate the associations between the quality of life and multimorbidity considering the effect of age, sex, sense of loneliness and WHOQOL. The significance was accepted for *p* < 0.05. All analyses were performed using SPSS V19.0 Statistical Software Package for Windows.

## 4. Results

All of the elderly patients studied lived with relatives and have different relation with family members and friends. The demographic characteristics of the sample including the distribution of the different diseases are described in Table 1. The presence of two or more pathologies in a single patient has been documented (Table 1). 

Moreover, the patients showed a medium (7%) to good (93%) level of self-functional sufficiency as evaluated by ADL (Activities of Daily Living).

Taking into consideration FACT-G total score and its subscales, they were reliable: FACT-G total score (Cronbach’s Alpha = 0.939), Physical-Well-Being (PWB) (Cronbach’s Alpha = 0.927), Social-Well-Being (SWB) (Cronbach’s Alpha = 0.927), Emotional-Well-Being (EWB) (Cronbach’s Alpha = 0.926), Functional-Well-Being (FWB) (Cronbach’s Alpha = 0.927).

So, we have considered these scales of quality of life for our analysis. In first we evaluated if some differences in quality of life dimensions between the different chronic diseases emerged. Differences between different pathologies did not emerge in quality of life but rather in the presence of multimorbidity comparing two with tree or more pathologies.

Significant differences in total score and in subscales of quality of life FACT-G (F = 1.689, *p* = 0.029), (PWB (F = 3.074, *p* = 0.003), SWB (F = 0.341, *p* = 0.003), EWB (F = 1.360, *p* = 0.012), FWB (F = 5.954, *p* < 0.001) emerged from the comparison between patients with two diseases and those with three and more than three pathologies. To be affected from three or more chronic diseases creates a negative blend on them (Table 2). This means that the quality of life decreases in elderly patients affected by of three or more diseases while those with one or two pathologies maintain a good quality of life, considering that we have studied self-sufficient and with intact cognitive abilities patients.

Then, we analyzed which conditions are associated with a sense of loneliness as subjective perception of relationships, social support, and demographic variables, and how much these factors may be considered as cofactors in addition to the multimorbidity in influencing the quality of life of these patients.

We proceeded to the elimination of possible multicollinearity of the variables (the study of correlations in Appendix A), from which the following variables emerged as variables to be included in the model: physical well-being, social well-being, emotional well-being, functional well-being, sense of loneliness, presence of social relationships, age, sex, multimorbidity. The only variable not resulting in the model is the educational level.

### Multiple Linear Regression

At this point it has finally come to the multiple linear regression models where the dimensions of quality of life FACT-G total score, PWB, SWB, EWB, and FWB, were considered as dependent variables for multimorbidity predictor also considering the correction for age, sex, sense of loneliness (4 categories), social support (2-category WHOQOL) (Table 3 and Appendix A).

The linear regression model with dependent variable FACT-G total score has a good adjusted R-square (0.630). It emerges that as the sense of loneliness increases (B = −0.285, *p* < 0.000), the FACT-G decreases by 0.2. Instead, as good social relationships increase, the PWB increases by 0.4 (B = 0.444, *p* < 0.000) (Table 3).

The linear regression model with dependent variable PWB has a good adjusted R-square (0.640). It emerges that in the presence of multimorbidity as the patient’s age increases (B = −0.024, *p* = 0.014), the PWB decreases by an average of 0.1, and as the sense of loneliness increases (B = −0.435, *p* < 0.000), the PWB decreases by 0.4. Instead, as good social relationships increase, the PWB increases by 0.4 (B = 0.463, *p* < 0.000) (Appendix A).

The linear regression model with social well-being (SWB) dependent variable has a not very high adapted R-square (0.474). It emerges that in the presence of multimorbidity, the SWB decreases by an average of 0.1, and as the sense of solitude increases, the SWB decreases by 0.4. Instead, as good social relationships increase, the SWB increases by 0.3.

With an increase in multimorbidity (B = −0.338, *p* = 0.009) SWB decreases by an average of 0.1, and as the sense of solitude increases (B = −0.401, *p* < 0.000), the social well-being decreases by 0.4. Instead, as good social relationships increase, the SWB increases by 0.4 (Appendix A).

The linear regression model with Emotional Well-Being (EWB) dependent variable has a adapted R-square (0.628). EWB decreases by an average of 0.1. It emerges that as the sense of solitude increases (B= −0.494, *p* < 0.000), the EWB decreases by 0.4. Instead, as good social relationships increase, the EWB increases by 0.4 (Appendix A).

The linear regression model with FWB dependent variable has an acceptable adjusted R-square (0.604). It turns out that in the presence of multimorbidity (B = −0.268, *p* = 0.012) the FWB decreases by 0.1. Moreover when the patient’s age increases (B = −0.023, *p* = 0.013) the FWB decreases by 0.1. As the sense of loneliness increases (B = −0.429, *p* < 0.000), the FWB decreases by 0.4.

## 5. Discussion

The results of the present study indicated that having a positive perception of the supportive social environment increases the quality of life of older people with physical illnesses, even in the presence of chronic disease and multimorbidity. The PWB, SWB, EWB, and FWB dimensions of QoL increase in the absence of a sense of loneliness in patients with multimorbidity, regardless of whether two or three to five pathologies are present. The elderly person maintains psycho-physical well-being even in the presence of many pathologies (multimorbidity) if he does not feel alone. The subjective perception of loneliness, as a lack of relationships perceived as meaningful and nourishing, is considered a risk of poor quality of life in patients diagnosed with chronic diseases and multimorbidity in the presence of self-sufficiency and intact cognitive abilities, as it was evaluated in the sample of this study.

Furthermore all of our patients lived with at least one family member and had daily social relationships and were therefore not isolated. So, experiencing a sense of loneliness is associated with bad PWB, SWB, EWB, and FWB. Moreover all of our patients have self-sufficiency.

That is, people with greater social participation and a positive feeling about the relationship have a lower risk of a poor quality of life even in the condition of suffering from multiple chronic diseases. Some authors affirm that social isolation increases the risk of being diagnosed with chronic illnesses; people with greater social participation have lower risk of suffering from multiple chronic diseases [35].

This finding may be in agreement with some suggestion emerged from our results. We agree with the consideration of Cantarero-Prieto [35] who affirms: “the risk linked to isolation, sense of loneliness in presence of relationships, together with the traditional one associated with lifestyles, should be considered in the development of new public policies in facing on the diseases especially in elderly people”. The study of Czaja [28], highlights the importance of social connectivity to well-being. Older adults with a small social network, and with greater physical and functional impairments may be particularly vulnerable to being socially isolated and lonely.

Some studies highlight the relationship between a sense of loneliness and frailty in a circle of mutual influence and recommend conducting screening and intervention programs to prevent frailty and loneliness in adults and older subjects [58]. Other studies have shown that loneliness is a cause of depression, which is in turn considered a threat to psycho-physical health [59].

The quality of social relationships should be considered a fundamental aspect of an individual’s lifestyle and well-being.

Our results add a further consideration: it is not mere social isolation as an objective dimension but feeling lonely, even in the presence of significant relationships, that influence the psycho-physical health of the elderly.

All of these considerations suggest, in agreement with Escourrou [60], that a purely physical approach to screening for multimorbidity and frailty syndrome should include a more holistic view of the older person in the context of the environment.

Thus, geriatric health workers are encouraged to evaluate the presence of social relationships and the subjective perception of these by the patient as a predictive dimension of good or bad adaptation to the condition of the disease and quality of life. The consequence is that interventions should be planned to stimulate the support provided by relatives and friends [31].

Our results are in agreement with extensive researches documenting the importance of subjective perception of quality of social relationships for promoting mental health and protecting against the development and progression of physical diseases [31]: social relationships, both quantity and quality, are a major contributing factor in lowering broad-based morbidity and mortality.

Age was another factor emerged from our study: PWB and FWB decreases with increasing age. This result confirms the results of other studies [4].

The scientific literature highlights that other factors that may contribute to the decline of health are the decrease in economic resources, illness, widowhood, and impaired mobility: all of these factors may result in increased risk for social isolation. In our sample, the patients were self-sufficient and without financial problems: we made this choice to avoid factors that would have influenced the results. However, it would also be interesting to evaluate the relation with these issues in further studies.

## 6. Limitations

First, all of the subjects in this study were affected by determined pathologies, and care is needed when extrapolating these results to other diseases. Second, a sampling bias was present in the data since all of the subjects attended one institution and thus were not representative of elderly patients affected by diseases and multimorbidity in general. Third, our results provide a snapshot of quality of life, sense of loneliness and other variables after diagnosis and during the treatment phase. The results in QoL symptoms may differ during the treatment phase or at other points in the disease journey. Quality of life may evolve naturally over time once patients begin being treated.

## 7. Conclusions

In conclusion we can make the following considerations:(1)The presence of a good perception of social relationships (absence of a sense of loneliness) can favor a good quality of life and a positive adaptation to the condition of chronic illness and multimorbidity and to medical treatment.(2)Patients with a sense of loneliness should be subjected to closer surveillance than those without such experiences.

Early identification of these patients may be necessary to provide mental health professionals with the opportunity for proactive intervention. In the context of palliative care and holistic approach, multidimensional and psychosocial interventions for improving the presence of a good perception of relationships (absence of a sense of loneliness) with patients and relatives may promote a good quality of life. Moreover, we hypothesize that psycho-social intervention in group may be more suitable in this context to promote natural defenses (for both patients and family members).

Furthermore, the psychotherapeutic intervention, which is rarely offered, should be prioritized under these circumstances (sense of loneliness affecting emotional state) [59].

Other studies will further explore new methods to support the specific needs of elderly patients, including interpersonal and emotional needs.

## Figures and Tables

**Table 1 ijerph-20-02615-t001:** Characteristics of sample.

Variables	N = 216 (%)
Age (years)	71.6 ± 5.5
Gender (female)	153 (70.8%)
Marital status	
Single	31 (14.4%)
Married	93 (43%)
Widower	80 (37%)
Separated/Divorced	12 (5.6%)
Educational level	
Elementary	24 (11.1%)
Middle school	63 (29.2%)
High	89 (41.2%)
University	40 (18.5%)
Type of relationship (Live with)	
Wife/Husband	93 (43.1%)
Son/daughter	89 (41.2%)
Other relatives	34 (15.7%)
Diseases	
Hypertension	167 (77.3%)
Cardiovascolar diseases	88 (40.7)
Diabetes	98 (45.4)
Cancer	72 (34.7)
Other pathologies	45 (20.8)
Multimorbidity (≥ 3 pathologies) Multimorbidity (2 pathologies)	137 (63.4%) 79 (36.5%)

Data are expressed by mean ± standard deviation for continuous variables, and by *n* (%) for categorical.

**Table 2 ijerph-20-02615-t002:** Independent samples’ test.

Variables	F	*p*
FACT-G total score	1.689	0.029
PWB	3.074	0.003
SWB	0.341	0.003
EWB	1.36	0.012
FWB	5.954	0.001

Physical well-being (PWB); social well-being (SWB); emotional well-being (EWB); functional well-being (FWB).

**Table 3 ijerph-20-02615-t003:** Coefficients (dependent variable: FACT_G total score).

			95.0% CI	
	B	*p*	Lower	Upper
(Constant)	3.087	<0.001	2.119	4.055
MULTIMORBIDITY	0.091	0.188	−0.045	0.227
AGE	−0.004	0.482	−0.017	0.008
GENDER	0.009	0.891	−0.116	0.134
SENSE OF LONELINESS	−0.285	<0.001	−0.365	−0.205
WHOQOL AGE	0.444	<0.001	0.285	0.604

Physical well-being (PWB).

## Data Availability

Appendix A can be obtained from the authors upon reasonable request.

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
