# Peer review of "Association between Sense of Loneliness and Quality of Life in Older Adults with Multimorbidity"

_ijerph, 2023, doi:10.3390/ijerph20032615_

Round 1

Reviewer 1 Report

The design of the study and methodology were chosen adequately to address the objectives. This study uses data from the Multidimensional Evaluation Form in Geriatrics project, a prospective observational study aimed at collecting data about the patterns and quality of prescriptions among older patients admitted to acute care wards of geriatric medicine located in Central Italy. For this analysis, patients affected by two or more (three to five) chronic health diseases were included: malign tumors, cardiovascular diseases, chronic pulmonary diseases, diabetes mellitus, hypertension. Methods Patients n. 162 with multimorbidity and living with family members. Tests: MMSE-Mini-Mental-State-Examination; ADL-Activities of Daily Living; Social Schedule: demographic variables; Loneliness Scale -de Jong Gierveld; Quality-of-Life-FACT-G;WHOQOL-BRIEF social relationships. Statistical analysis: Multivariate Regression Analysis.

The results clearly presented. The patients with three or more diseases have worse dimensions of QoL Physical-well-being(p=.003), Social well-being (p=.003), Emotional-well-being(p=.012), Functional-well-being(p<.001), than those with two. The presence of a good perception of relations (absence of a sense of loneliness) and its link with good quality of life can help a positive adaptation to the condition of chronic illness and multimorbidity. The absence of a sense of the dimension of loneliness can be an indicator of patients with a good adaptation to disease conditions related to medical treatment. 2) patients with a sense of loneliness should be subjected to closer surveillance than those without these experience.

Conclusion. Multidimensional and psychosocial interventions for improving the presence of a good perception of relations (absence of a sense of loneliness) with patients and relatives may promote a good quality of life favoring a positive adaptation to the condition of chronic illness and multimorbidity.

The limitation: a sampling bias was present in the data because all the subjects attended one institution and thus were not representative of elderly patients affected by diseases and multimorbidity in general.

A strong part of the study was good methodology and the design of the study. The weaker part of the study was that the study was not representative.

The manuscript is suitable for printing. It is recommended to print without corrections.

Author Response

On behalf of other Authors, I thank the Reviewer for his/her positive commentson the study. We agree with the Reviewer comment "The weaker part of the study was that the study is not representative".

Unfortunaltely, given the complexity of the multidimensional record data, clinical data and psychosocial variables  to be collected, it was difficult to involve  other institutes or universities, also due to the lack of funds. The INRCA_IRCCS as a Geriatric Institute provides a standard assessment of data expected within the multidimensional evaluation. For this reason was possible to carry out this study.

Reviewer 2 Report

This study is one of the needs of medical centers

Author Response

Response to Reviewer 2

On behalf of other Authors I thank the Reviewer for his/her positive comment on the study.  Following the reviewer's suggestion the conclusions have been improved, relevant references were included and the conclusions were improved (highlighted in the text).

Reviewer 3 Report

Dear Editors and authors,

Thank you for inviting me to review this manuscript. This is an exploration to understand the factors associated with the sense of loneliness and quality of life in a population of older adults with multimorbidity, that shows potential. However, there are opportunities for improvement, as is outlined below.

MAIN CHANGES

3. Statistical analysis

The authors did not describe if any test was performed to determine whether the data followed a normal distribution. Data needs to follow a normal distribution, so that parametric tests can be used, this is the case of Pearson´s Coefficient.

If data distribution was not assessed, it must be done.

It would be necessary to display a table with the Results of Pearson´s Coefficient between the studied variables in order to show a full picture of the association between them and quality of life in adults with multimorbidity.

4.Results

The linear regression model using the total score of FACT_G as the dependent variable, should be performed and displayed in the manuscript.

At the same time, it could be advisable to provide as supplementary material the tables displaying the results of the linear regression models for PWB, SWB, EWB and FWB.

5.Discussion

Lines 326-330 and lines 344-346: Letter size is bigger than in the rest of the manuscript.

The discussion section has a lot of room for improvement, there is very little discussion of the results of the present study comparing them with the existing literature and the conclusions of the study are expressed in the discussion section.

It may be worth to consider including a “limitations” section in the “discussion” and having a separated “conclusions” section.

MINOR CHANGES

Title, Author´s names and Affiliations:

Verify if you are following the journal´s format guidelines.

Keywords

Besides “Elderly” all the keywords of the manuscript also appear in the title, it should be worth using synonyms since it would increase manuscript´s readability.

1.Introduction

Line 82: May something be missing in one of the two brackets? ([ ]). 

Lines 95-97: Check the veracity of this sentence, as there are some reviews and metanalysis addressing loneliness and mortality in older adults. Some examples are DOI: 10.1097/PSY.0000000000001151 and DOI: 10.1007/s10433-022-00740-z .

2.1.Participants

Expressing sample numbers with letters may not be adequate, using numbers would increase the readability of the manuscript.

Line 123: Check if it should be “>=3” instead of “>3”

Lines 123-124: There is discontinuity between the lines.

Line 137: One of the points at the end of the line may need to be deleted.

Line 143: "refusing to participate" needs to be mentioned as an exclusion criteria?

2.2. Measures

The description of the tests used has room for improvement.

References:

Authors should cite in Vancouver style, if you are referencing an author´s work and we cite his/her name, the citation number needs to appear to. This is not done in line 203 and line 307. Please verify this in the journal´s format guidelines.  

Author Response

Response to Reviewer 3

On behalf of other authors , I thank the Reviewer for his/her comments and suggestions.

Statistical analysis

Response: We have checked the normality  and so we have used parametric tests. Now you can see this new sentense in statistical analysis session: "All continuous variables were checked for normaqlity of distribution by Kolmogorov-Smirnov and the result was that all of them were normally distributed".

Response: We added the results of Pearson's correlations in the Table 1 of Supplementary Material (Table S1).

Results

We change Table 3 -regression model (in which FACT_G total score is the dependent variable) and we also added its Crombach's alpha and its independent sample's test in Table 2.

About other linear regression model with PWB, SWB, EWB, FWB as dependent variables we added them in Tables S2, S£,S$ and S% (Supplementary Material).

Discussion

Response:

The size of the letters of Lines 326-330 and 344-346 have been reduced as in the rest of the text.

The discussion section has been expanded (highlighted in yellow).

The "Limitations" section has been placed in the "Discussion" and the "Conclusion" section has been separated.

Round 2

Reviewer 3 Report

Dear Editors and authors,

In first place, thank you for listening to my recommendations. I really appreciate the effort made to improve the quality of the manuscript, which you have successfully achieved. Congratulations on your work. However, there is still some room for improvement, as it is outlined below.

Discussion

As the topic of the manuscript is a field where many research can be found in the same and in similar populations, I suggest that you include some more references in this section in order to give a deeper comparison of your result with the available body of evidence in the topic.

In the remaining parts of the manuscript great quality changes have been made. Congratulations for your great work.  

Thank you for your time.

Author Response

To Reviewer 3

Thanks so much for your comments and suggestions.

We have added the references (#47, 48, 60-highlighted in yellow) in the introduction and discussion.

Discussion has been improved:

Some studies highlight the relationship between a sense of loneliness and frailty in a circle of mutual influence and recommend conducting screening and intervention programs to prevent frailty and loneliness in adults and the elderly. Other studies have shown that loneliness is a cause of depression which is considered a threat to psycho-physical health.